# Quality of the Pellets Obtained with Wood and Cutting Residues of Stone Pine (*Pinus pinea* L.)

Manuel Fernández [ID], Raúl Tapias, Victoriano Camacho and Joaquín Alaejos *[ID]

Higher Technical School of Engineering, Campus of "El Carmen", University of Huelva, 21007 Huelva, Spain;
manuel.fernandez@dcaf.uhu.es (M.F.); rtapias@dcaf.uhu.es (R.T.); victoriano.camacho@dcaf.uhu.es (V.C.)
* Correspondence: joaquin.alaejos@dcaf.uhu.es

**Abstract:** The use of wood and residual biomass from forestry works is a $CO_2$ emission-neutral source of energy that also contributes to reducing the risk of spreading forest fires, especially under Mediterranean climate. The forest stands of stone pine (*Pinus pinea* L.) occupies about 0.7 million hectares in the Mediterranean basin. In this study, the commercial quality of the pellets manufactured from different types of cutting residues (needles and thin branches, medium branches and bark), as well as wood from trunks and thick branches, was assessed. It was concluded that with the exclusive use of residual biomass it is not possible to obtain pellets of high commercial quality, useful for residential or industrial use. However, the highest quality pellets could be obtained by combining them with stone pine debarked wood, but in a certain proportion that differs depending on the type of residue (around 15% for bark, 30% for medium branches and less than 15% for needles and thin branches). It is recommended to take advantage of both the thick wood (trunk + thick branches) and a proportion of medium branches and bark, while in the case of needles and thinnest branches it would be more convenient to leave them in the forest for their incorporation into the soil, given their high nutrients concentration and their low quality for energetic use. The results found support a greater valorization of the biomass obtained in the stone pine fellings. In the future it will be necessary to study which is the most appropriate logistics of the silvicultural works to be able to conveniently apply the results of this study.

**Keywords:** *Pinus pinea* L.; forest residues; quality pellet; fire forest; bioenergy; mechanical durability; ash content

## 1. Introduction

Nowadays, the use of renewable energy is increasingly necessary. On the one hand, because it is a clean and non-polluting energy and, on the other hand, in order to achieve energetic independence with respect to the ever scarcer fossil fuels. This causes many countries to have to import them on a large scale from producing countries [1].

Biomass is a renewable energy that has been used since ancient times [2]. Today, it is increasingly used with the advantage of adding value to the waste generated in the tasks derived from forestry and agricultural management. In addition, the energetic use of residues from sustainable forest exploitation is a viable way of reducing combustibility in forest stands. For this reason, the use of forest biomass has a double advantage, both producing neutral energy in $CO_2$ emissions, and contributing to the prevention of forest fires, a serious environmental problem, especially in countries with a Mediterranean climate. [3–5]. This is motivated by the severe weather conditions in these areas with high temperatures and extensive summer drought.

The global annual production of biomass covers about 14% of the global energetic demand and 35% of the energetic demand of developing countries [6]. Bioenergy is still currently the most widely used renewable energy worldwide and its demand will continue to grow [7].

The low density and the high moisture content, as well as the consequent difficulties for transport and storage, are the main problems related to the energetic use of biomass [8,9]. Faced with these drawbacks, pelletizing has clear advantages over other biomass fuels that have not been densified. The higher energetic density of the pellets and their low moisture content, around 10%, result in lower transportation costs and higher energetic conversion efficiency. These properties also allow for greater long-term storage capacity [10].

The use of pellets is expanding more and more. In 2019, the world production of pellets was 40.5 million tons, which represented an increase of 125% compared to seven years ago [11]. In the EU, the consumption of pellets in 2020 reached 19.3 million tons, which represented an increase of 5% compared to the consumption of the previous year [12].

Pine wood is one of the most used types of biomass for the manufacture of pellets, due to its properties such as a high resin content, its unit density and being a softwood. The great worldwide distribution of the genus *Pinus*, as well as its low ash content and its high calorific value, superior to that of other woody and non-woody species, make it very suitable for this use [13]. Numerous works have studied the properties of the pellets of different pine species. Scots pine (*Pinus syslvestris* L.) wood, either alone [14–20] or mixed with other species, such as Norway spruce (*Picea abies* (L.) H. Karst) [21–25] or beech (*Fagus sylvatica* L.) [26], has been extensively studied. Other pine species have also been assessed, such as Monterey pine (*Pinus radiata* D. Don) [27,28]; Eldar pine (*Pinus eldarica* Medw) [27]; Aleppo pine (*Pinus halepensis* Mill.) [29]; maritime pine (*Pinus pinaster* Ait.) [29–31]; Calabrian pine (*Pinus brutia* Ten.) [32]; Lodgepole pine (*Pinus contorta* Douglas ex Loudon) [32,33]; Eastern white pine (*Pinus strobus* L.) [34]; etc.

The stone pine (*Pinus pinea* L.) is a species that occupies an area of about 0.7 million hectares in countries surrounding the Mediterranean basin [35]. The country with the largest area of this species is Spain with 470,000 ha, followed by Portugal (80,000 ha), Turkey (50,000 ha) and Italy (40,000 ha) [36]. This pine has also been successfully introduced to Argentina, South Africa, the United States [37]; Chile [38]; Algeria, Tunisia and Morocco [39].

As far as we know, there are few studies on the manufacture of pellets from stone pine. Some of them have been carried out in Italy [40]. In Portugal, the combustion process in domestic boilers with stone pine pellets has been analyzed, comparing them with those of other species [41]. There is also a study showing the calorific value of 100 different biofuels, including stone pine [42]. Fernández et al. [43] determine the physico-mechanical properties of pellets made from debarked *Pinus pinea* wood from trunks and thick branches.

In addition, the risks of forest fires are increasing in countries with a Mediterranean climate, due to the effects of climate change and the accumulation of plant biomass in forest areas. This increased risk of forest fires is especially significant in stone pine forests [44]. Therefore, any activity that involves the use and energetic valorization of wood and residual biomass of this species, derived from silvicultural work, will contribute to reducing the combustibility of these forests and improving the prevention of forest fires.

Establishing whether it is possible to obtain quality pellets from stone pine cutting residues will allow a higher value use of these forest stands. This would entail being able to pay for forestry works recommended for fire prevention and contribute to obtaining an economic and environmental benefit in the management of these masses. To achieve this main objective in the present study, a series of trials are established leading to the following tasks:

- Manufacture and characterization of pellets obtained from the different types of biomass originated by forestry works in *Pinus pinea* stands.
- Determine the optimal operating ranges such as pressure, particle size and moisture to obtain stone pine wood pellets.
- Evaluate the properties of pellets made with a mixture of wood and residual pine biomass.

## 2. Materials and Methods

The raw material for the manufacture of pellets has been wood and woody waste from forestry work for fire prevention in April 2021, carried out in the province of Huelva (La Rábida, 37°12′04″ N, 6°55′11″ W), southwest of Spain. The works consisted of a moderate thinning of the stone pine stand, eliminating 30 to 50% of the existing trees and reaching an optimal density of the stand [45]. In the forest that is the object of the study, the final density after felling, as it is an adult pine forest, was 450 trees/ha. The mean diameter, measured at 1.30 m from the ground, of the cut trees was 25.1 cm and the mean height was 8.82 m. The stone pine is the dominant native forest species in Huelva, occupying 28% of the wooded forest area of this province, with more than 100,000 hectares [46] (Figure 1).

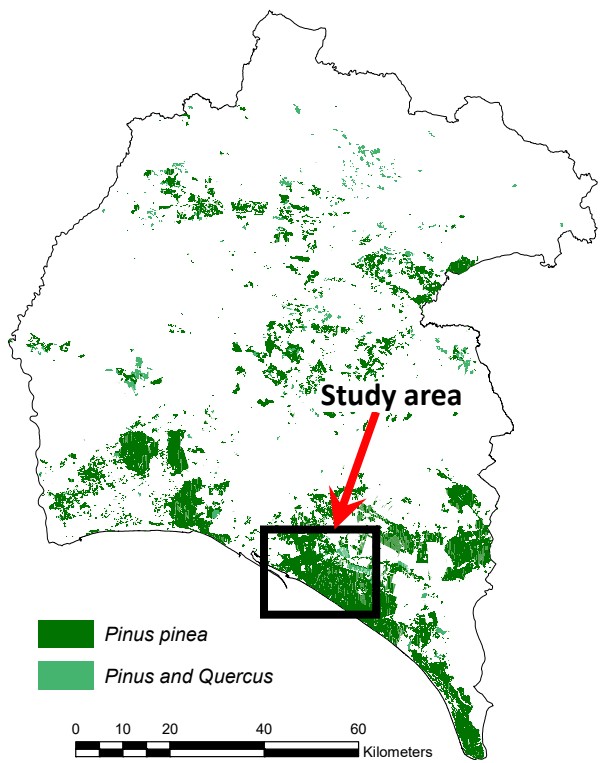

**Figure 1.** Distribution of stone pine in the province of Huelva and location of the study area.

### 2.1. Assay with Residual Biomass of Pine

The raw material (biomass) was obtained from 8 trees, randomly selected from among all those that were cut, but with the condition of being separated by more than 50 m from one another. The residual biomass (medium and fine branches, bark and leaves) was pooled into 2 groups, 4 trees/group; while for thick debarked wood, 4 groups were made (2 trees/group). According to Ruiz-Peinado et al. [47] the average distribution of the dry biomass of *Pinus pinea* trees with the size of those felled here is: 41.6% in the stem, of which 10.4% corresponds to the bark and 31.2% to the wood; 22.7% in the thick branches (>7.5 cm in diameter); 13.3% in the middle branches; 15.7% in the thin branches (<2.5 cm in diameter); and 6.6% in the needles.

Firstly, with the aim of adding value to the residual biomass from the *Pinus pinea* pine forest, pellets were made with three types of this biomass:

- Needles and twigs (NTW).
- Medium branches with bark (diameter between 2.5 and 7.5 cm, –MBR–)
- Bark of logs of more than 20 cm in diameter at breast height (BARK).

At the same time, pellets of debarked thick pine wood (DTTB) were also made, by mixing 65% from trunks of 20–35 cm in diameter and 35% from thick branches of >7.5 cm in diameter, which usually produces pellets of the highest commercial quality (e.g., ENplus

A1) [48]. The chemical and physical-mechanical properties of all of them were assessed and compared with the ISO 17225-2:2021 standard, which establishes the methods of analysis and criteria of commercial quality of the pellets [49].

The four types of biomass were ground to sawdust with a knife mill equipped with a 5 mm sieve and then allowed to air dry. The particle size distribution is shown in Table S3. The moisture content of the sawdust at the time of manufacture was around 13% (12.8 ± 0.3%). The pellets were manufactured using 4 types of dies, differentiated by the length of the die channels (18, 20, 22 and 24 mm), whose diameter was 6 mm. The difference in length implies a difference in the pressure necessary for the biomass to pass through the channels and come out compressed at the opposite end. The pelleting pressures were calculated for each die according to the equation of Holm et al. [16] for pine. The resulting pressures were 90.2, 120.3, 159.4, and 210.4 MPa, respectively. Dies with a shorter (16 mm) or longer (26 mm) length were not used because, with the equipment used (PLT-400, Smartec®, Villar San Costanzo, Italy), either the sawdust was not compacted into pellets and it came out of the pelletizer just as it was (16 mm), or it was too compacted and got clogged or burned inside the die (26 mm). No additive was added. Therefore, the experimental design was as follows:

$$4 \text{ dies} \times 4 \text{ biomass types} \times 2\text{–}4 \text{ repetitions} = 40 \text{ pellet batches}$$

The parameters measured on the manufactured pellets, according to the ISO 17225-2 standard, were:

- Moisture content (%).
- Bulk density ($kg/m^3$), both with moisture just after manufacturing and based on dry mass (0% moisture).
- Mechanical durability (%).
- Length and diameter (mm).
- Ash content (%) on a dry mass basis.
- N, S, Cl, C and H content (%) on a dry mass basis (105 °C), as well as low calorific value (LCV, kWh/kg) both based on dry mass and with the moisture just after manufacturing. Kjedahl method (auto-analyzer Bran+Luebbe®, Mod. AIII; Norderstedt, Germany) for N; ICP-OES (Thermo Jarrell Ash Corporation; Franklin, MA, USA) for K and P; total extraction with $HNO_3$ and photometry (HI 83200, Hanna® Instruments; Woonsocket, RI, USA) for Cl; Elemental analyzer (Thermo Scientific™ FLASH 2000; Chino, CA, USA) for S, C and H; automatic isoperibol calorimeter (Parr 6300®; Parr Instrument Company; Moline, IL, USA) for LCV.

  It was also determined:

- Pellet particle density ($kg/m^3$), both with moisture just after being manufactured and on a dry mass basis.
- Process yield (% *w/w*): proportion of pellets greater than 3.15 mm in length just after manufacturing.

The data were analyzed using a two-factor ANOVA, die (*D*) and type of biomass (*B*), considered as fixed, and the interaction Die × Biomass was included in the model ($y_{ijk} = \mu + D_i + B_j + (D \times B)_{ij} + \varepsilon_{ijk}$). Previously, the data normality and homoscedasticity checks were performed. Differences were considered significant at $p < 0.05$. The differences between the means were analyzed using the Bonferroni test. Likewise, the existing correlations between the measured parameters were analyzed by means of Pearson's linear correlation (r) and by means of non-linear correlations.

### 2.2. Factorial Assay with Debarked Thick Pine Wood

Secondly, in order to optimize the manufacture of pine biomass pellets using debarked coarse wood alone or mixed with the residual biomass obtained from forestry work in the pine forest, it was considered necessary to previously determine the optimal manufacturing conditions wood pellets with the pelletizing machine used. With the results obtained here,

in the subsequent test we proceeded to analyze and quantify the proportion of residual biomass that could be mixed with debarked thick wood to obtain pellets of the highest commercial quality (ENplus A1). To do this, based on the results of the previous test (Section 2.1), a new test was designed only with thick debarked *Pinus pinea* wood. The variables studied were three: particle size of the sawdust, pressure exerted on the palletizing die by the rollers and sawdust moisture.

The raw material used was the same thick debarked *Pinus pinea* wood (DTTB) described in the first assay, which had been passed through a 5.0 mm sieve. Three of the 4 groups of that sawdust were taken and then each one was divided into three sublots, two of which were passed through the mill again, but this time with sieves of different openings, one 2.0 mm and the other 3.5 mm. Therefore, material with three particle sizes was obtained for each group (named by the sieve used, 2.0, 3.5 and 5.0 mm). All sublots of sawdust were allowed to air dry, at room temperature and under cover, and the moisture stabilized on this occasion at 14%. In turn, each one of the sublots was again divided into three other sublots, to which the moisture was adjusted to 11.5, 14.0 and 16.5%. These moisture contents were chosen since previous experiences discouraged the use of sawdust with lower (due to lack of compression) or higher (excess steam production) moisture content. The pellets were manufactured using the same four types of dies as in the previous assay (18, 20, 22 and 24 mm in length), all of them 6 mm in diameter. Dies of greater or lesser length than these were not used for the reasons mentioned above. The measured parameters were the same as in the previous assay and the experimental design was as follows:

$$4 \text{ dies} \times 3 \text{ sieves} \times 3 \text{ moistures} \times 2\text{–}3 \text{ repetitions} = 88 \text{ pellet batches}$$

The pellet mill used does not allow setting a fixed working temperature value. However, it is refrigerated by air from a compressor that allows manual regulation, at all times, of the flow of fresh air that reaches the die, being able to maintain its temperature in a range of $\pm 5\,°C$. Temperature measurements were carried out by means of a double laser infrared thermometer (Scantemp 490, Dostmann electronic GmbH; Reicholzheim, Germany) pointing directly at the sawdust-free die from above the feed hopper, every 4–5 min, from a distance of about 40 cm.

Data were analyzed using a three-way ANOVA: length of the die channels ($D$), sieve or particle size ($S$) and sawdust moisture ($Ms$), and all possible interactions were included in the model ($y_{ijkl} = \mu + D_i + S_j + Ms_k + (D \times S)_{ij} + (D \times Ms)_{ik} + (S \times Ms)_{jk} + (D \times S \times Ms)_{ijk} + \varepsilon_{ijkl}$). Previously, the data normality and homoscedasticity checks were performed. Differences were considered significant at $p < 0.05$. The pairwise comparisons of the means were analyzed using the Bonferroni test. Likewise, the existing correlations between the measured parameters were analyzed by means of Pearson's linear correlation (r) and by means of non-linear correlations. Finally, a multiple regression analysis of the measured properties was also carried out, based on the 3 variables involved (sieve width [$S$, mm], pressure inside the die channels [$P$, MPa] and sawdust moisture [$Ms$, %]).

### 2.3. Mixtures of Residual Biomass with Debarked Pine Wood

Thirdly and lastly, once the results of the previous tests had been obtained (in the first one the properties of the pellets made with different fractions of stone pine biomass; in the second one the best operating ranges for the manufacture of pellets of wood of this species with the pelletizer used), the design of this test was established. In this case, the raw material was a mixture of thick debarked pine wood with a fraction of each of the three types of residual biomass studied (NTW, BARK, MBR).

Taking into account the ash, N, S and Cl content of the *Pinus pinea* biomass used (Tables 1, S1 and S2), the maximum percentages of residual biomass that could be mixed with the debarked wood without undermining the chemical commercial quality for domestic use of the latter (e.g., ENplus A1) were determined. These would be of the order of 15% for bark, 15% for NTW and up to 40% for MBR on a dry mass basis.

**Table 1.** Mean values of the mineral content of the different types of biomass obtained from *Pinus pinea* dry mass.

| Type of Biomass | N (%) | P (%) | K (%) | S (%) | Cl (%) | H (%) | C (%) | LCV [1] (kWh/kg) | LCV [2] (kWh/kg) |
|---|---|---|---|---|---|---|---|---|---|
| BARK | 0.38 | 0.020 | 0.14 | 0.060 | 0.025 | 5.7 | 52.1 | 5.09 | 4.52 |
| DTTB | 0.16 | 0.010 | 0.10 | 0.025 | 0.015 | 6.3 | 50.8 | 5.13 | 4.86 |
| MBR | 0.32 | 0.015 | 0.15 | 0.040 | 0.020 | 6.2 | 51.5 | 5.18 | 4.78 |
| NTW | 1.05 | 0.072 | 0.47 | 0.115 | 0.030 | 6.4 | 50.4 | 5.12 | 4.53 |

Low calorific value (LCV) has been determined based on dry mass [1] and with the moisture content just after manufacturing [2].

In a first approximation, three mixtures were tested in order to evaluate the physical-mechanical properties of the pellets. According to the results of the previous tests, the moisture of the raw material (mixture of sawdust) was adjusted to 15%, and the 22 mm die was used. The resulting particle size of the 5 mm sieve was chosen as it is more common in the industry, is less costly in time and energy than sawdust passed through smaller sieves, and can provide the same quality of pellets as those with finer particles. The proportions of the selected mixtures were the following:

- 85% DTTB + 15% BARK (*w/w*)
- 70% DTTB + 30% MBR (*w/w*)
- 85% DTTB + 15% NTW (*w/w*)
- Control (100% DTTB)

In the case of MBR, the proportion was lowered from the maximum possible (40%) to 30%, since this is approximately the relative proportion by weight in a tree of these two fractions to be mixed (13.3%/41.6% = 0.32). In addition, in the first trial MBR presented worse pellet quality than DTTB, so it is not justified to increase the proportion in the mixture above 30%.

The data were statistically analyzed using a one-way ANOVA (raw material mix –Mz–), according to the model $y_{ij} = \mu + Mz_i + \varepsilon_{ij}$. Differences were considered significant for $p < 0.05$. The differences between the means were analyzed using the Tukey test. The measured parameters were the same as in the previous assays and the experimental design was as follows:

$$1 \text{ die} \times 1 \text{ sieve} \times 1 \text{ moisture} \times 4 \text{ mixtures} \times 3 \text{ repetitions} = 12 \text{ pellet batches}$$

**3. Results**

*3.1. Assay with Residual Biomass of Pine*

The chemical and physical properties of the raw materials used in the manufacture of the pellets are shown in Table 1.

The results revealed significant differences between types of biomass, between dies or manufacturing pressure, as well as for the interaction of both factors (Die × Biomass type) for all the parameters studied (Tables 2 and 3). Table S4 shows the values of the properties of the different pellets made.

The ash content did not differ significantly between dies or for the Die × Biomass type interaction ($p > 0.07$), but it did differ between types of biomass ($p < 0.001$): 2.76 ± 0.04% (NTW), 2.19 ± 0.04% (BARK), 1.23 ± 0.04% (MBR) y 0.32 ± 0.02% (DTTB), being the ranking NTW > BARK > MBR > DTTB. The pellets diameter varied between 6.04 ± 0.01 and 6.10 ± 0.01 mm, so within the range allowed in the quality standards, but without differentiating between types of biomass or die used.

As can be seen in Table 3, the significance of the Die × Biomass interactions means that we must look at both the differences between the different levels of the main factors and the interaction. Figure 2 shows four of the measured parameters.

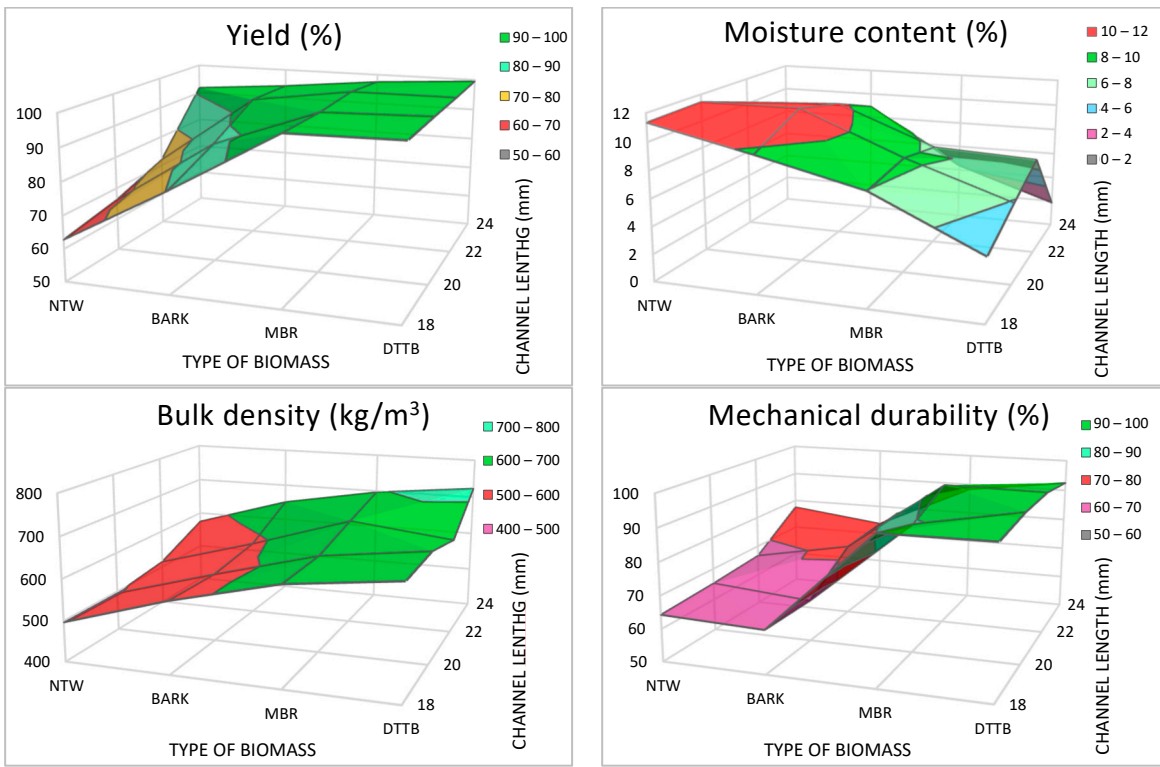

**Figure 2.** Mean values of yield, moisture, bulk density and durability of pellets made with 4 types of biomass (NTW, Bark, MBR, DTTB) and 4 dies (channel length of 18, 20, 22 and 24 mm).

**Table 2.** Pellets properties (mean ± σ) with different dies and types of stone pine biomass.

| | Yield (%) | Pellet Moisture (%) | Durability (%) | Bulk Density (kg/m³) | Pellet Length (cm) | Particle Density (kg/m³) |
|---|---|---|---|---|---|---|
| **Channel length of the die (mm)** | | | | | | |
| 18 | 88.2 ± 15.4 (a) | 7.61 ± 2.91 (b) | 81.8 ± 15.6 (a) | 609.6 ± 70.1 (a) | 1.48 ± 0.39 (b) | 1190.5 ± 105.5 (a) |
| 20 | 90.1 ± 13.6 (b) | 8.62 ± 2.68 (c) | 81.6 ± 15.3 (a) | 609.1 ± 71.7 (a) | 1.38 ± 0.40 (ab) | 1218.2 ± 67.8 (ab) |
| 22 | 92.3 ± 11.2 (c) | 8.50 ± 1.76 (bc) | 84.9 ± 13.3 (b) | 609.7 ± 60.1 (a) | 1.01 ± 0.38 (ab) | 1169.6 ± 79.7 (ab) |
| 24 | 96.5 ± 3.2 (d) | 5.14 ± 3.35 (a) | 84.9 ± 8.4 (b) | 673.7 ± 59.4 (b) | 1.25 ± 0.18 (a) | 1278.0 ± 59.2 (b) |
| **Biomass type** | | | | | | |
| NTW | 73.2 ± 12.0 (a) | 10.41 ± 1.17 (c) | 68.4 ± 6.0 (a) | 520.2 ± 34.8 (a) | 1.09 ± 0.13 (a) | 1111.5 ± 65.3 (a) |
| BARK | 89.0 ± 6.4 (b) | 10.14 ± 1.04 (c) | 68.1 ± 5.5 (a) | 595.5 ± 35.2 (b) | 0.88 ± 0.09 (b) | 1153.7 ± 74.5 (a) |
| MBR | 98.6 ± 0.9 (c) | 7.08 ± 1.72 (b) | 91.6 ± 3.1 (b) | 660.2 ± 26.1 (c) | 1.61 ± 0.22 (c) | 1248.1 ± 37.5 (b) |
| DTTB | 99.5 ± 0.2 (c) | 4.86 ± 2.22 (a) | 94.2 ± 1.8 (c) | 675.8 ± 34.4 (c) | 1.66 ± 0.19 (c) | 1278.4 ± 44.1 (b) |
| **TOTAL MEAN** | 92.0 ± 11.8 | 7.47 ± 2.99 | 83.3 ± 13.0 | 625.5 ± 69.0 | 1.38 ± 0.37 | 1214.0 ± 87.2 |

Different letters in the same column indicate significant differences (Bonferroni test), according to *p*-values shown in Table 3, after the ANOVA test considering significant differences at $p < 0.05$.

The properties measured in the pellets were significantly correlated with each other (Table S5 and Figure 3), highlighting that:

- The ash content of the raw material was significantly correlated with the bulk density, durability, production yield and length of the pellets.
- In this study, a yield greater than 95% was only achieved for ash contents below 1.3%; the bulk density exceeded 600 kg/m³ for ash contents below 2.18%; and the mechanical durability exceeded 95% only if the ashes were below 1.21%.
- Yield exceeded 95% only when mechanical durability surpassed 88%, which only happened for MBR and DTTB. Durability values above 96% were only achieved by DTTB. Pellets with a durability greater than 95% were not obtained if the bulk density

did not exceed 600 kg/m$^3$. Similarly, the 95% yield was only achieved when the bulk density exceeded 600 kg/m$^3$.

**Table 3.** Significance level obtained in the factorial ANOVA applied to the pellets made with 4 types of biomass (NTW, BARK, MBR, DTTB) and 4 dies (18, 20, 22 and 24 mm).

| Property | *p* (Significance Level) | | |
| --- | --- | --- | --- |
| | **Biomass Type** | **Die** | **Die × Biomass** |
| Yield (%) | <0.001 | <0.001 | <0.001 |
| Pellet moisture (%) | <0.001 | <0.001 | 0.001 |
| Durability (%) | <0.001 | <0.001 | <0.001 |
| Bulk density (kg/m$^3$) | <0.001 | <0.001 | 0.003 |
| Bulk density 0% moisture (kg/m$^3$) | <0.001 | <0.001 | 0.003 |
| Pellet length (cm) | <0.001 | <0.001 | <0.001 |
| Particle density (kg/m$^3$) | <0.001 | <0.001 | 0.004 |
| Particle density 0% moisture (kg/m$^3$) | <0.001 | <0.001 | 0.010 |

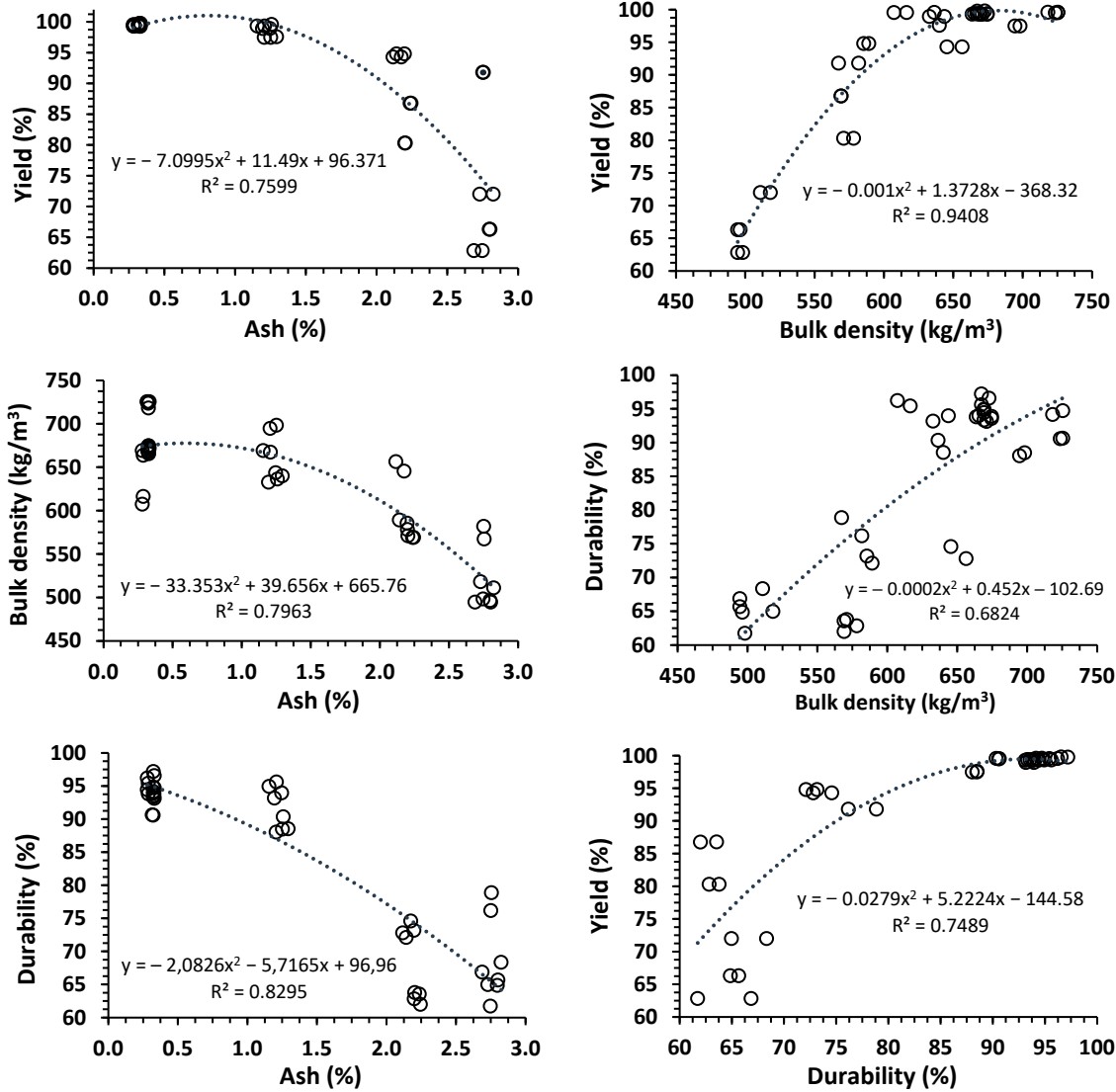

**Figure 3.** Correlations between parameters for pellets made with 4 types of biomass (NTW, BARK, MBR, DTTB) and 4 dies (18, 20, 22 and 24 mm).

### 3.2. Factorial Assay with Debarked Thick Pine Wood

The results obtained revealed significant differences between sieves (particle size) and dies (pressure), and some for sawdust moisture, as well as for the *Die × Sieve* and *Sieve × Moisture* interaction (Tables 4 and 5). Table S6 shows the values of the properties of the different pellets made. The interaction *Die × Moisture* and the triple *Sieve × Die × Moisture* were not significant ($0.074 < p < 0.924$), except for durability ($p < 0.001$ and $p = 0.002$, respectively). The ash content coincided with that of the previous test $0.32 \pm 0.02\%$ (DTTB). The diameter of the pellets obtained varied between $6.02 \pm 0.01$ and $6.07 \pm 0.01$ mm, therefore, within the range admitted in the quality standards.

**Table 4.** Mean values ($\pm\sigma$) of the properties of stone pine wood pellets with different dies (channel length 18, 20, 22 and 24 mm), moisture content (MOI, 1 = 11.5%, 2 = 14%, 3 = 16.5%) and particle sizes (SF = 2 mm; SM = 3.5 mm; SC = 5 mm).

| | Yield (%) | Pellet Moisture (%) | Durability (%) | Bulk Density (kg/m$^3$) | Pellet Length (cm) | Particle Density (kg/m$^3$) |
|---|---|---|---|---|---|---|
| **Channel length of the die (mm)** | | | | | | |
| 18 | 99.4 ± 0.3 (a) | 6.64 ± 2.98 (b) | 93.7 ± 2.0 (a) | 631.5 ± 53.5 (a) | 1.86 ± 0.18 (b) | 1273.6 ± 48.5 (a) |
| 20 | 99.7 ± 0.3 (b) | 7.02 ± 1.50 (b) | 96.3 ± 1.6 (c) | 654.6 ± 28.7 (a) | 1.89 ± 0.16 (b) | 1267.2 ± 22.9 (a) |
| 22 | 99.5 ± 0.2 (ab) | 7.48 ± 1.61 (b) | 95.6 ± 1.3 (bc) | 627.7 ± 36.4 (a) | 1.82 ± 0.19 (b) | 1224.2 ± 26.0 (a) |
| 24 | 99.6 ± 0.1 (ab) | 3.05 ± 1.62 (a) | 94.5 ± 1.9 (b) | 706.1 ± 22.9 (b) | 1.40 ± 0.21 (a) | 1298.7 ± 35.0 (b) |
| **Pellets moisture content (%)** | | | | | | |
| MOI1 | 99.5 ± 0.2 (a) | 5.10 ± 1.82 (a) | 94.5 ± 1.9 (a) | 668.8 ± 27.4 (b) | 1.73 ± 0.23 (a) | 1271.2 ± 34.2 (b) |
| MOI2 | 99.6 ± 0.3 (a) | 6.16 ± 2.77 (b) | 95.4 ± 1.7 (b) | 655.5 ± 45.2 (ab) | 1.76 ± 0.31 (a) | 1268.7 ± 47.3 (ab) |
| MOI3 | 99.6 ± 0.2 (a) | 6.81 ± 2.98 (b) | 95.0 ± 2.3 (ab) | 640.4 ± 64.5 (a) | 1.72 ± 0.26 (a) | 1255.9 ± 44.6 (a) |
| **Particle size (mm)** | | | | | | |
| SF | 99.5 ± 0.2 (b) | 5.24 ± 2.41 (b) | 94.4 ± 2.0 (b) | 669.0 ± 41.5 (a) | 1.70 ± 0.25 (b) | 1273.3 ± 43.3 (a) |
| SM | 99.5 ± 0.2 (a) | 5.53 ± 2.58 (a) | 94.5 ± 1.8 (a) | 666.0 ± 40.6 (b) | 1.67 ± 0.28 (a) | 1276.4 ± 42.5 (b) |
| SC | 99.8 ± 0.1 (a) | 7.71 ± 2.37 (a) | 96.5 ± 1.4 (a) | 622.5 ± 52.5 (b) | 1.91 ± 0.20 (ab) | 1241.0 ± 35.7 (b) |

Different letters in the same column indicate significant differences (Bonferroni test), according to *p*-values shown in Table 5, after the ANOVA test considering significant differences at $p < 0.05$.

**Table 5.** Significance level obtained in the factorial ANOVA applied to the pellets made with debarked thick pine wood with 3 types of sawdust moisture (11.5, 14.0 and 16.5%), 3 sieves (2.0, 3.5 and 5.0 mm) and 4 dies (18, 20, 22 and 24 mm).

| Property | *p* (Significance Level) | | | | |
|---|---|---|---|---|---|
| | *Sieve* | *Die* | *Moisture* | **S × D** | **S × M** |
| Yield (%) | <0.001 | <0.001 | 0.542 | 0.641 | 0.579 |
| Pellet moisture (%) | <0.001 | <0.001 | <0.001 | 0.002 | <0.001 |
| Durability (%) | <0.001 | <0.001 | <0.001 | <0.001 | <0.001 |
| Bulk density (kg/m$^3$) | <0.001 | <0.001 | <0.001 | 0.001 | <0.001 |
| Bulk density 0% moisture (kg/m$^3$) | <0.001 | <0.001 | <0.001 | 0.002 | <0.001 |
| Pellet length (cm) | 0.016 | 0.001 | 0.358 | 0.654 | 0.372 |
| Particle density (kg/m$^3$) | <0.001 | <0.001 | 0.041 | 0.001 | 0.002 |
| Particle density 0% moisture (kg/m$^3$) | <0.001 | <0.001 | <0.001 | <0.001 | <0.001 |

As can be seen in Table 5, the significance of some *Sieve × Die* and *Sieve × Moisture* interactions means that we should look at both the differences between the different levels of the main factors and the interactions (Table S7 and Figures 4 and 5).

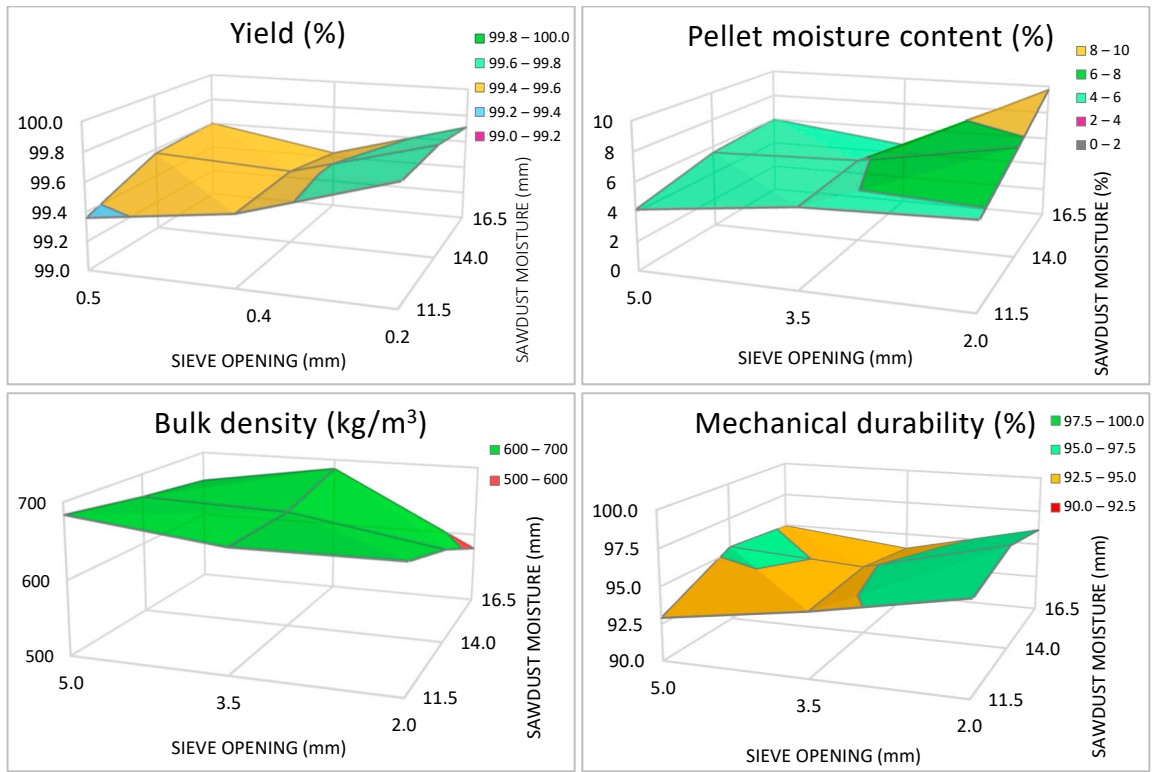

**Figure 4.** Mean values of yield, moisture, bulk density and durability of pellets made from debarked thick pine wood (DTTB) milled to 3 particle sizes (2.0, 3.5 and 5.0 mm sieves) and 3 sawdust moistures (11.5, 14.0 and 16.5%).

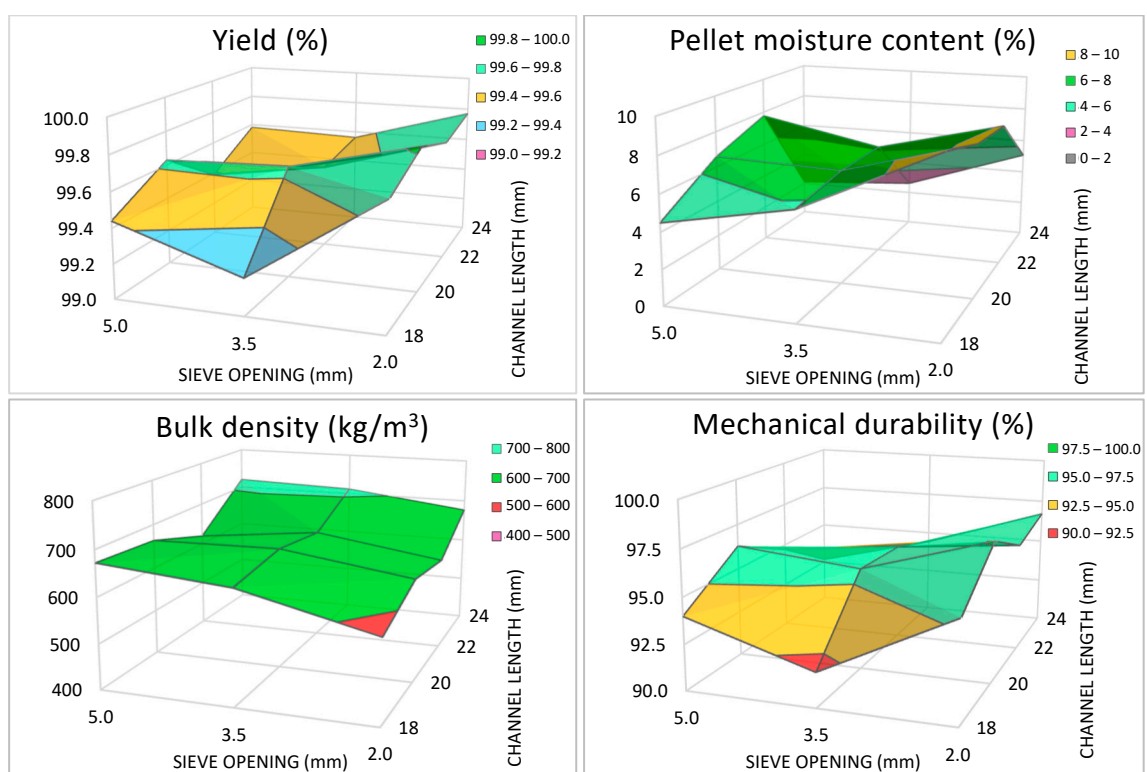

**Figure 5.** Mean values of yield, moisture, bulk density and durability of pellets made from debarked thick pine wood (DTTB) milled to 3 particle sizes (2.0, 3.5 and 5.0 mm sieves) and 4 dies (channel lengths of 18, 20, 22 and 24 mm).

The properties measured in the pellets were correlated with each other (Table S7 and Figure 6), highlighting that:

- In this study, a bulk density greater than 600 kg/m$^3$ was only achieved for pellet moisture below 9%, while durability greater than 97.5% was achieved for pellet moisture between 5 and 10%. Therefore, this optimal range of 5%–9% corresponded to pellets with a mean length of 1.5–2.3 cm, very suitable for commercial use.
- Yield exceeded 96.6% only when durability was greater than 96%, and 99.7% for durability of 97.5%. These are values of fairly optimal yield in the process of manufacturing commercial grade pellets.

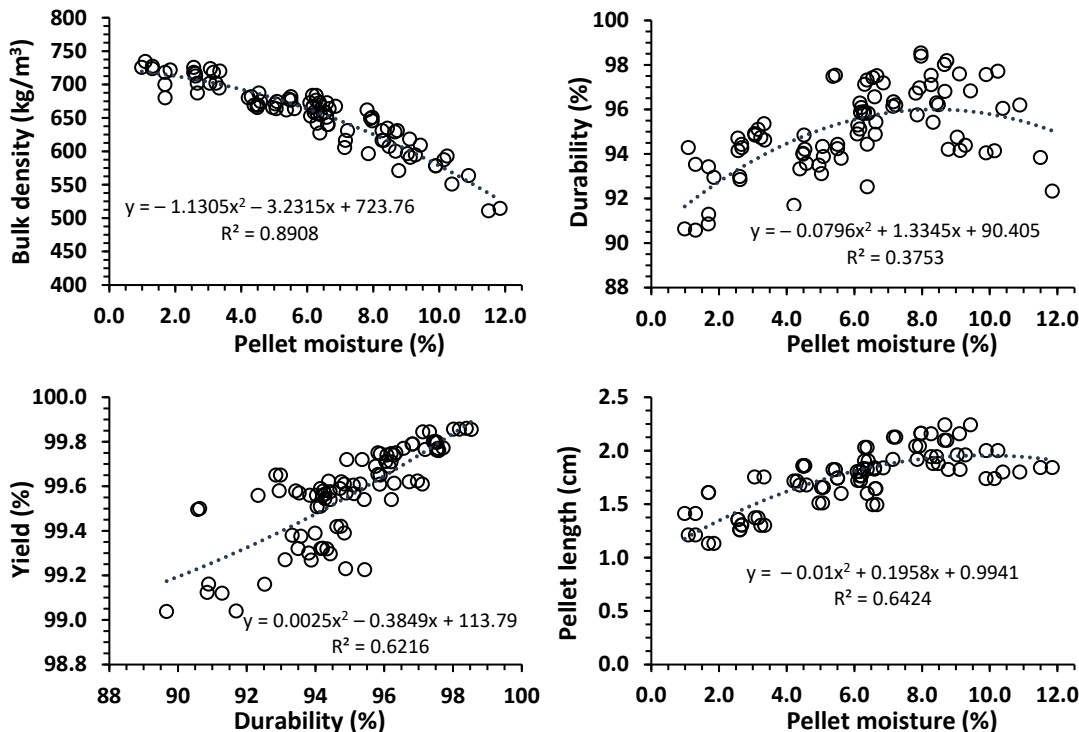

**Figure 6.** Correlations between parameters of pellets made from debarked thick pine wood, 3 particle sizes (2.0, 3.5 and 5.0 mm sieves), 3 moisture content in sawdust (11.5, 14.0 and 16.5%) and 4 dies (18, 20, 22 and 24 mm).

On the other hand, from the multiple regression analysis of the properties measured as a function of the 3 variables involved (sieve opening [*S*, mm], pressure in the die [*P*, MPa] and sawdust moisture [*Ms*, %]), we can highlight these three relationships:

- Pellet moisture (%) = $8.600 - 0.029\,P - 0.871\,S + 0.341\,Ms$ R = 0.669, R$^2$corr = 0.428, $p < 0.001$
- Bulk density (kg/m$^3$) = $601.003 + 0.542\,P + 15.793\,S - 5.582\,Ms$ R = 0.666, R$^2$corr = 0.424, $p < 0.001$
- Pellet length (cm) = $2.545 - 0.004\,P - 0.065\,S$ R = 0.707, R$^2$corr = 0.489, $p < 0.001$

The regressions for the other measured parameters (durability, yield, particle density) were not significant and/or presented a coefficient of determination (R$^2$) below 0.3.

Therefore, the variance explained by the multiple regression models, in the best of cases, was close to 50%, leaving 51%–68% unexplained, which could be due to the experimental error itself or to changes in manufacturing temperature conditions. The latter could be regulated within a range of values, but it could not be set at a specific value in this pelletizing equipment. The working temperature was 95–105 °C for the 18 mm die, 100–115 °C for the 20–22 mm die and 115–125 °C for the 24 mm die.

### 3.3. Mixtures of Residual Biomass with Debarked Pine Wood

The results obtained are summarized in Table 6. With the range of pellet moisture obtained (5.7%–7.9%), the real lower calorific value, with the moisture content at the exit of the pelletizer, oscillated between 4.77 and 4.69 kWh/kg, always higher than the minimum limit established by the marketing regulations for top quality pellets.

**Table 6.** Mean values ($\pm\sigma$) of the physical-mechanical properties of the residual biomass mixtures after their transformation to pellets. Different letters in the same row indicate significant differences ($p < 0.001$, Tukey's test).

| Property | Biomass Mix Type | | | |
| --- | --- | --- | --- | --- |
| | 85% DTTB 15% NTW | 85% DTTB 15% BARK | 70% DTTB 30% MBR | 100% DTTB |
| Yield (%) | >99.5% (no significant differences, $p = 0.120$) | | | |
| Pellet moisture (%) | 7.9 $\pm$ 0.4 b | 6.5 $\pm$ 0.4 ab | 5.6 $\pm$ 0.4 a | 5.7 $\pm$ 0.3 a |
| Durability (%) | 97.0 $\pm$ 0.4 a | 98.1 $\pm$ 0.4 a | 98.6 $\pm$ 0.4 a | 98.0 $\pm$ 0.3 a |
| Bulk density (kg/m$^3$) | 596.2 $\pm$ 4.0 a | 617.2 $\pm$ 4.0 b | 644.0 $\pm$ 4.0 c | 644.2 $\pm$ 2.8 c |
| Pellet length mm) | 20.5 $\pm$ 4.0 (no significant differences, $p = 0.488$) | | | |
| Particle density (kg/m$^3$) | 1142 $\pm$ 11 a | 1196 $\pm$ 11 b | 1192 $\pm$ 11 b | 1252 $\pm$ 8 c |
| Pellets classes | Does not comply | A1 | A1 | A1 |

## 4. Discussion

The manufacture of quality pellets requires threshold values for their properties. In the case of pellets for domestic and industrial use, the ISO 17225-2 standard (Tables S1 and S2) establishes the following as minimum quality limits: moisture $\leq$ 10%; bulk density $\geq$ 600 kg/m$^3$; durability $\geq$ 96.5% and ashes $\leq$ 2.0%. In the case of the pellets obtained with different kinds of biomass from stone pine, it can be verified:

- The fine branches and needles (NTW) do not meet the minimum criteria for commercial quality with any of the matrices used, so that, in addition to their poor chemical quality (high ash and mineral element content), they are mechanically unsuitable for pelletizing. The latter limits the proportion of this type of biomass that could be mixed with thick debarked pine wood if maximum commercial quality is to be achieved. The longest die used in this trial (24 mm) achieved acceptable levels for pellet yield and moisture content, but not for mechanical durability or bulk density. It is the only biomass type for which pellets could be produced with a longer die (26 mm, 276.7 MPa), but still failed to reach acceptable durability levels.
- Pine bark only showed acceptable values for yield, moisture and bulk density with the 24 mm die, but also did not reach acceptable levels of mechanical durability. The high ash and mineral content also limits its pelleting quality.
- As expected, biomass from debarked logs and thick branches (DTTB) showed the best properties. In some cases, the medium branches with bark were close to the DTTB values, without significant differences, but in general with poorer quality. Although the differences between the 4 dies were not very pronounced for these four properties, the 20 and 22 mm dies showed better results. For the 24 mm die, the DTTB moisture content was too low (<2%).
- The ash content of a plant tissue is closely related to the concentration of mineral nutrients (N, P, K, etc.). Non-woody or less woody tissues, such as leaves, the cambium-phloem region and part of the bark usually have higher nutrient concentrations than wood, as shown in Table 1. Likewise, it has been shown that the physical-mechanical characteristics of the pellets (e.g., bulk density, mechanical durability) are closely related to their composition, mainly lignin, which is more abundant in woody than non-woody tissues [8,23,31]. For this reason, this relationship between ash content and physical-mechanical properties of the different types of pine biomass used has been revealed and quantified, highlighting a reduction in yield and quality

of the pellets with the increase in ash content. The positive relationship between yield, mechanical durability and bulk density is more logical, since if the operating conditions are adequate, the sawdust particles bind well, resulting in compact pellets and less production of unbound fine elements.

It was also observed that the quality of pellets obtained with the a priori best quality biomass (DTTB), although acceptable, did not reach the optimum values obtained in the only known test with stone pine wood [40]. This is possibly due to the low moisture content of the sawdust used, which is why it was proposed to modify the moisture content for the next test and continue to maintain the same matrices used in this one.

In the second trial where pellets were made only from debarked wood using a factorial design with different moisture content, die channel length and particle size, it was observed that:

- No sample made with sawdust moisture content of 11.5% met all the minimum commercial quality criteria, reducing the choice in this study to moisture contents between 14.0 and 16.5%.
- No sample made with the 18 mm die met all the minimum commercial quality criteria, which narrows the choice in this study to dies between 20 and 24 mm, corresponding to pelleting pressures of 120 to 210 MPa.
- The smallest particle size (2.0 mm sieve) with a sawdust moisture content of 16.5% usually presents problems of excess moisture in pellets and low bulk density for shorter dies (18 to 22 mm). This restricts the use of this particle size to 14 % moisture for 20–22 mm matrices, or to 24 mm matrices if 16.5% moisture is used.
- The maximum temperature range used for the tests as a whole, 95–125 °C, is within a range in which the effect of temperature on the properties of the pellets is not too high [10,50], at least for pines [17]. However, it could have partially affected the differences obtained between the factors tested.
- With the range of pellet moisture content obtained (2.6%–11.7%) the actual lower calorific value at pellet moisture content at the pelletizer outlet ranged from 4.95 to 4.47 kWh/kg. This indicates that the combination of short die and sawdust with high moisture content can result in pellets that do not meet the marketing standard for LCV. The moisture content of the pellets will depend on factors such as sawdust moisture, pelletizing temperature, pressure, etc. An excess of moisture in the pellets decreases the mechanical durability and the bulk density, perhaps due to a high amount of water in the sawdust that hinders the formation of hydrogen bonds during the bonding of the particles [17]. The calorific value increases the lower the moisture, however a moisture content too low gave rise to shorter and more compact pellets that increased the bulk density too much, almost exceeding the recommended commercial limits (750 kg/m$^3$), and made them more brittle, decreasing their mechanical durability. For this reason, the most appropriate range of pellets moisture was between 5% and 9%.
- The intermediate and larger particle sizes (3.5 and 5.0 mm sieves, respectively) presented mechanical durability problems with the 24 mm matrix.
- Of the 88 individual samples produced, 8 (11%) meet the highest commercial quality, ENplus A1, with mechanical durability values > 98.0%, pellet moisture < 10% and bulk density > 600 kg/m$^3$, and a yield of more than 98.5%. The pellets with these requirements were: in the case of the finest particle size (2 mm sieve) those obtained with a 20 mm die and a moisture content of 16.5%. For 5 mm particle size, quality pellets are obtained with 24 mm die and 14.0 to 16.5% moisture, as well as with 20 mm matrix and 14.0% particle moisture. These would be our reference ranges for moisture, particle size and die for the pelletizer used, which will serve as a reference for the next test on mixtures of residual biomass with wood.

Mechanical durability is a quality parameter that depends on both the raw material and the pelletizing process. In this test, in addition to having obtained various types of pellets that comply with the maximum commercial quality (ENplus A1), others obtained

with other operating conditions are very close to meeting these requirements, some of them complying with the quality limits immediately below the maximum, ENplusA2. Possibly with an industrial pelletizer with more power than the one used in the test, it would have been possible for these pellets to reach the maximum quality. Even in the third test shown in this study, it was possible to achieve this quality with greater ease. The maximum mechanical durability value obtained was 98.5%. This value is higher than the one obtained by Ilari et al. [40] for pine wood pellets, 95.9. Although in this work the pelleting conditions are different, so they do not allow a good comparison with the results of the present study.

From the results obtained in the 3rd test where pellets are obtained from wood and stone pine residues, it can be seen that the pellets obtained with the mixtures of bark (15%) and medium branches (30%) continue to meet the minimum requirements to be marketed with ENplus A1 quality. On the other hand, the mixture of wood with fine branches and needles (at 15%) would not comply, since the bulk density, although by a very small margin, remains below the limit of 600 kg/m$^3$. Therefore, for this last mixture, the percentage of residues would have to be reduced and, possibly, between 10 and 14% would be sufficient.

In future works, it is proposed to make pellets with mixtures of pine wood together with various types of logging residues, determining the appropriate proportions to obtain quality pellets. With these results, it would also be necessary to study the sequence of operations in the silvicultural treatments in order to be able to conveniently use the residues by means of whole-tree systems. These systems would be combined with other systems that would allow part of the biomass, such as needles and twigs, to be shredded and incorporated into the soil. For example, according to Table 1 and Section 2.1, by not removing the needles and fine branches (NTW) of the cut trees from the forest stand, which represent only 22.3% of the aboveground dry biomass harvested, the forest preserves the 55% of N and K, and the 66% of P, contained in said aboveground dry biomass.

## 5. Conclusions

-   Residual biomass from pine forest operations can be varied and have significant differences ($p < 0.001$) not only for chemical but also for physical-mechanical properties, depending on whether they are branches, bark, twigs or leaves. For. Example, for all the tested channel length of dies as a whole and 13% of sawdust moisture, DTTB, MBR, BARK and NTW averaged, respectively: 4.9, 7.1, 10.1 and 10.4% for pellet moisture; 676, 660, 596 and 520 kg/m$^3$ for bulk density; and 94.2, 91.6, 68.1 and 68.4% for mechanical durability.
-   Residual pine biomass can be valorized for energetic use, but its valorization after transformation into pellets would not reach commercial quality standards for use in domestic or industrial boilers by itself. An exception could be the industrial use of biomass from medium branches (2.5–7.5 cm diameter) with bark.
-   Residual biomass could be mixed with biomass from coarse wood (debarked trunk and thick branches) up to a limit in weight ratio and still obtain pellets of the highest commercial quality. This limit will vary depending on the type of residual biomass, around 15% for bark (BARK), 30% for medium branches (MBR) and less than 15% for needles and fine branches (NTW), not only because of the limitations in chemical composition but also because of the devaluation of the physical-mechanical properties of the pellets.
-   More tests could be carried out to more accurately determine the maximum percentages of residues, more specifically in the case of middle branches (MBR) and bark. However, the values will be very close to those established in the last test, 30% and 15%, respectively, since the values of some properties, especially mechanical durability, are very close to the minimums admitted for an ENPlus A1 quality.
-   It would be appropriate to plan silvicultural treatments to take advantage of both the trunk and medium branches (MBR) and bark (77.7% of the aboveground dry biomass). In this way, the thin branches and needles (the remaining 22.3%) would remain in the

forest, which would contribute to the recycling of nutrients (55%–65% of N, K and P) and their incorporation into the soil.

- It must be considered that the nutrient content of the thin branches and needles is the highest of the entire tree, for example 10 times more N concentration than in the wood. Also, due to their higher N and S content on combustion, they release more $NO_2$ and $SO_2$ than biomass from other parts of the plant. This, together with the low physical-mechanical quality of the pellets obtained with these fine branches and needles ($\leq$575 kg/m$^3$ bulk density, and $\leq$77.5% mechanical durability), as can be deduced from the tests carried out in this work, makes it advisable to leave these residues on the forest.
- In any case, if the lightest residues, or even others, are left in the forest, they should always be shredded or chipped in situ to avoid the risk of fire and facilitate their incorporation into the soil.

**Supplementary Materials:** The following supporting information can be downloaded at: https://www.mdpi.com/article/10.3390/f14051011/s1. Table S1. Specification of wood pellet classes for commercial and residential uses (ISO 17225-2); Table S2. Specification of the classes of wood pellets for industrial use. (ISO 17225-2); Table S3. Granulometric distribution of the sawdust used in the tests for each type of biomass and sieve; Table S4. Pellets properties with different dies and types of stone pine biomass; Table S5. Pearson correlation coefficients among the characteristics of the pellet samples of pine residues; Table S6. Properties of stone pine wood pellets with different dies, moisture content and particle sizes; Table S7. Pearson correlation coefficients among the characteristics of the pellet samples of debarked pine trunk.

**Author Contributions:** The dedication of each one of the authors in the elaboration of this work has been the following: Conceptualization, J.A. and M.F.; methodology, J.A., M.F., V.C. and R.T., software, M.F. and R.T.; formal analysis, J.A. and M.F.; investigation, J.A. and M.F.; resources; writing—original draft preparation, J.A., M.F. and V.C.; writing—review and editing, J.A. and M.F.; supervision, M.F.; funding acquisition, J.A. All authors have read and agreed to the published version of the manuscript.

**Funding:** This study was financed by Projects: "Iberian Center for Research and Forest Firefighting", CILIFO (0753_CILIFO_5_E) and "Strengthening of cross-border systems for the prevention and extinction of forest fires and improvement of resources for rural employment generation post COVID-19", FIREPOCTEP (0756_FIREPOCTEP_6_E). Both are Projects co-financed by the European Regional Development Fund (ERDF), within the framework of the Interreg V A Spain—Portugal Program (POCTEP) 2014–2020.

**Data Availability Statement:** All the data is available on request from the corresponding author.

**Conflicts of Interest:** The authors declare no conflict of interest.

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
