# Peer review of "Quality of the Pellets Obtained with Wood and Cutting Residues of Stone Pine (Pinus pinea L.)"

_forests, doi:10.3390/f14051011_

Round 1

Reviewer 1 Report

This work is related to the use of Pinus pinea biomass for production of solid biofuels in the form of pellets. Of course, the subject is not so novel but still useful. The authors seem to have made a lot of work but the paper presents some very important scientific weaknesses for which I suggest its rejection.

First of all, according to the authors, the manuscript had 4 aims. In my opinion, most of them have not been properly addressed since the methods that the authors chose to carry out did not lead to results that could help them accomplish their goals. The most representative example is the fourth aim: “to assess the possibility of using stone pine residues as a measure to prevent forest fires”. There is no assessment of forest fire risk in this manuscript. Furthermore, in relation to this aim the authors claim that “the prevention of forest fires is really a consequence of the previous three aims, since the valorization of pine residues will make it possible to reduce the combustibility of these forest stands and therefore the risk of fires.” In my opinion this is a clear example of conclusion drawn without the support of related findings.

Regarding aim #2 the authors stated that they would “Determine the ideal operating conditions such as pressure, particle size and moisture to obtain stone pine wood pellets”. Nevertheless, there are no results or drawn conclusions that show the effects of pressure, particle size or particle MC on the quality of the produced pellets.

Finally, regarding the properties of pellet mixtures (aim #3), the authors used a confusing experimental setup which in my opinion did not provide a clear view of the effect of each separate raw material or their mixtures on the quality of the produced pellets. For this reason, the interpretation of the results and the drawing of conclusions were very difficult.

Another important weakness of this work is that, even though the authors report that the various production batches were carried out in different temperatures in the range of 95-125oC they seem to not take these important differences in account. It is well known that among densification parameters, temperature belongs to the most important ones since it influences glass transition temperature of constituents. The authors seem to disregard this.

Other weak points of this work:

The authors use terminology that is not known or not common in this field (eg: “aerial dry mass”, “energetic properties”)

134-135 the used model for this determination was made for pinus sylvestris but the authors used it for another species.

268-279 and Fig 3, There is no interpretation provided for the acquired results. What is the reason for which the increasing ash content results to reduced yield, bulk density and mechanical durability? How can these results be explained?

Fig. 6. The same with the above comment

Fig. 2 is confusing because it includes quantitative values.

334-335: The authors should have reported the method for determination of temperature and take the densification temperatures into account in their analysis.

340-345 confusing paragraph, makes no sense.

348-349 which criteria do the authors refer to?

There are conflicts in the basic idea of the manuscript. An example is that the authors claim that the removal of tree biomass from the forest could be a good measure for the prevention of forest fires and at the same time they state that “branches and needles would remain in the forest, which would contribute to the recycling of nutrients and their incorporation into the soil.“ Isn’t this biomass responsible for forest fires? What is the truth and what is the authors contribution towards finding the truth?

It also detected that the manuscript has been submitted in the section: “Forest Ecology and Management” which is not related to the scope of the work.

Author Response

We appreciate the suggestions and comments you have made to our article. In accordance with these, we have proceeded to make the modifications detailed in the attached file.

Reviewer 2 Report

The manuscript is interesting for an alternative approach to the selection of parts of the Pinus pinea L. energy tree to obtain high-quality pellets. To do this, according to the authors, it is necessary to mix the parts of the tree in a certain proportion.

There are some recommendations for improvement:

1. It is appropriate to add two sentences at the end of the abstract that answer the questions: "What do the results mean in practice? And what remains unresolved?"

2. Please explain the poor quality of all the figures. The quality must be improved

3. For a more complete demonstration of all the advantages of the experiment, it is desirable to supplement the Introduction section with information about the practical application of the results, the formulation of the problem (why is it necessary to study this issue?), the purpose of the research (indicating the tasks to be solved). I recommend clearly defining the number of tasks in the introduction, and briefly and clearly summarizing their achievement in conclusion.

4. To increase the reader's interest, it is advisable to disclose future research at the end of the Discussion section.

5. What is ALS in keywords? This abbreviation is not disclosed anywhere in the text.

6. In conclusion, it is necessary to remove the inappropriate statement and provide specific numerical values of the data obtained with an indication of the p level.

7. It is desirable to specify DOI in the references.

8. The Data Availability Statement section is missing. Where can I see the results (datasets) of these studies? You must provide a link to them.

Author Response

(The authors gave the same response as above.)

Reviewer 3 Report

Requirements

1.      Title: I think the more accuracy title would be: “Quality of the pellets obtained with wood and cutting residues of stone pine (Pinus pinea L.)”. In this way, it is more clearly observed that a combination of wood and leftovers will be used.

2.      Line 8. More appropriated: “forestry works”.

3.      Line 22. Other keywords could be added: quality (quality pellet), (thin) pine branch; mechanical durability, ash content, etc.

4.      Lines 25-27. This phrase must be divided into two parts, so that it can be better understood. Also "energy independence" should be changed to "energetic independence". Change the same word in whole paper. For instance, at line 44, “energetic density”.

5.      Line 34.” CO2 emissions,”. Pay attention to subscripted: “CO2”.

6.      Line 36. Do not forget the point “.” At the end of sentences. Please explain, why the Mediterranean basin is prone to fires (probably due to the high temperatures and autoignition of dry forest vegetation).

7.      Lines 44-46. Make this phrase more understanding. Could be divided.  

8.      Line 52. You must explain why the pine is a good specie for pellet production (compactify, unit density, resin content, expansion coefficient of sawdust, etc).

9.      Lines 69-70. Phrase “A few of them in Italy [40].” is in the air. Please make the connection with the previous sentence, or put its verb.

10.   Line 83. “works”. Pinus pinea, in italics. See also line 110.

11.   Line 84. Change and replace the “ideal” word with “optimal” one.

12.   Lines 87-93. This phrase must be inserted in the method chapter, after it is adapted accordingly.

13.   Line 101. Please be more specific and say how many trees are left per hectare of forest (usually between 2000-5000 trees/ha).

14.   Line 105-106. Please add. “…randomly selected from 105 among all those that were cut”.

15.   Fig. 1. The data in the figure are in Spanish. Please translate them into English, and make the picture clearer (it's a little blurry).

16.   Line 125. Title of the standard can be erased. (Solid biofuels - Fuel specifications and classes - Part 125 2: Graded wood pellets,. It is enough to be included in the bibliography and cited in parentheses within the research.

17.   Lines 137-138. Please add to each piece of equipment, next to the company, the city and country of its origin. Make the same for entire paper.

18.   Linse 172-173. Please make the connection between the two separated sentences.

19.    Lines 178-179. Please change. “…pressure exerted on the pelletizing die by roles”.

20.   Table 1. It must be explained in the table what LHV (probably Low heat value) represents, especially since it is a different terminology than the international one; HCV-high calorific value, LCV-low calorific value.

21.   Lines 246-247. The expression “The chemical and energetic properties of the raw materials used in the manufacture of the pellets are shown in Table 1” is repetitive to lines 240-241. Please remove it.

22.   Table 2. Please explain inside of this table what represent “σ, sigma” character and “Mean”. Pay attention the next aspects: 1. “σ, sigma” is assigned to standard deviation of lots and “s” is assigned to standard deviation of specimen; 2. “mean ± sigma” is assigned to a confidence interval of about 77%, “mean±2 sigma” is assigned to a confidence interval of 95%, and “mean±3 sigma” is assigned to a confidence interval of 99.7% (from statistical theories).3. Keep the same font “Palatino linotype” inside of table.

23.   Fig. 3. If possible, you should have the same (writing) font “Palatino linotype” in all figures.

24.   Line 329. Pay attention to superscript “(R2)”. Table 4. Pay attention to superscript “(kg/m3)”. Line 448. “600 kg/m3”. Line 478 : NO2 and SO2

25.   Lines 353, 367. “15% humidity”. Change to “15% moisture content”.

26.   Line 441. The next sentence must be improved, in order to be more understanding. “Although the working and pelletising conditions do not make 441 this work and the present one comparable”.

27.   Line 451. In the conclusions chapter, you should give up the introduction, and each conclusion should have its own paragraph. Short them.

28.   Lines 482-490. It was not possible to access the 7 tables S1, S2,… and S7. I recommend that these tables be included in the work, or otherwise no more reference should be made to them.

Author Response

(The authors gave the same response as above.)
